# Ecto-NOX Disulfide-Thiol Exchanger 2 (ENOX2/tNOX) Is a Potential Prognostic Marker in Primary Malignant Melanoma and May Serve as a Therapeutic Target

**DOI:** 10.3390/ijms252111853

**Published:** 2024-11-04

**Authors:** Matti Böcker, Eftychia Chatziioannou, Heike Niessner, Constanze Hirn, Christian Busch, Kristian Ikenberg, Hubert Kalbacher, Rupert Handgretinger, Tobias Sinnberg

**Affiliations:** 1Division of Dermatooncology, Department of Dermatology, University of Tuebingen, Liebermeisterstraße 25, 72076 Tuebingen, Germanyeftychia.chatziioannou@med.uni-tuebingen.de (E.C.); heike.niessner@med.uni-tuebingen.de (H.N.); constanze.hirn@med.uni-tuebingen.de (C.H.); 2Department of Urology and Pediatric Urology, University Hospital of Ulm, Albert-Einstein-Allee 23, 89081 Ulm, Germany; 3Department of Nutritional Biochemistry, Institute of Nutritional Sciences, University of Hohenheim, Garbenstraße 30, 70599 Stuttgart, Germany; 4Cluster of Excellence iFIT (EXC 2180) “Image Guided and Functionally Instructed Tumor Therapies”, Eberhard Karls University of Tuebingen, 72076 Tuebingen, Germany; 5Dermatologie zum Delfin, Stadthausstraße 12, 8400 Winterthur, Switzerland; christian-busch@hotmail.com; 6Institute of Clinical Pathology, University Hospital Zuerich, Schmelzbergstraße 12, 8091 Zuerich, Switzerland; kristian.ikenberg@usz.ch; 7Institute of Clinical Anatomy and Cell Analysis, University of Tuebingen, Elfriede-Aulhorn-Straße 8, 72076 Tuebingen, Germany; kalbacher@uni-tuebingen.de; 8Department of General Pediatrics, Hematology and Oncology, University Children’s Hospital Tuebingen, Hoppe-Seyler-Straße 1, 72076 Tuebingen, Germany; rupert.handgretinger@med.uni-tuebingen.de; 9Department of Dermatology, Venereology and Allergology, Charité-Universitaetsmedizin Berlin, Charitéplatz 1, 10117 Berlin, Germany

**Keywords:** malignant melanoma, ecto-NOX disulfide-thiol exchanger 2, ENOX2, tNOX, phenoxodiol, vemurafenib, biomarker

## Abstract

With an increasing incidence of malignant melanoma, new prognostic biomarkers for clinical decision making have become more important. In this study, we evaluated the role of ecto-NOX disulfide-thiol exchanger 2 (ENOX2/tNOX), a cancer- and growth-associated protein, in the prognosis and therapy of primary malignant melanoma. We conducted a tissue microarray analysis of immunohistochemical ENOX2 protein expression and The Cancer Genome Atlas (TCGA) *ENOX2* RNA expression analysis, as well as viability assays and Western blots of melanoma cell lines treated with the ENOX2 inhibitor phenoxodiol (PXD) and BRAF inhibitor (BRAFi) vemurafenib. We discovered that high ENOX2 expression is associated with decreased overall (OS), disease-specific (DSS) and metastasis-free survival (MFS) in primary melanoma (PM) and a reduction in electronic tumor-infiltrating lymphocytes (eTILs). A gradual rise in ENOX2 expression was found with an increase in malignant potential from benign nevi (BNs) via PMs to melanoma metastases (MMs), as well as with an increasing tumor thickness and stage. These results highlight the important role of ENOX2 in cancer growth, progression and metastasis. The ENOX2 expression was not limited to malignant cell lines but could also be found in keratinocytes, fibroblasts and melanocytes. The viability of melanoma cell lines could be inhibited by PXD. A reduced induction of phospho-AKT under PXD could prevent the development of acquired BRAFi resistance. In conclusion, ENOX2 may serve as a potential prognostic marker and therapeutic target in malignant melanoma.

## 1. Introduction

Despite a stabilization of incidence in certain subgroups, the general incidence of melanoma continues to increase with, in the United States alone, an estimated 100,640 new cases of melanoma and an additional 99,700 new cases of in situ melanoma in 2024. Although the mortality rate is decreasing due to better treatment options and early detection, 8290 deaths from melanoma are predicted [1]. This highlights the importance of the disease as well as the need for further risk stratification strategies and improved treatment options.

Approximately 45–50% of all melanomas harbor activating mutations in the *BRAF* gene of the MAPK signaling pathway [2,3]. Over 90% of these *BRAF* mutations affect codon 600 and involve an exchange of the amino acid valine (*BRAF*^V600^) [4]. Targeted therapies with BRAF inhibitors (BRAFis) like Vemurafenib selectively inhibit the constitutively activated BRAF^V600^ protein and can lead to a prolonged median progression-free survival of approximately 5–6 months [5,6,7]. Nevertheless, acquired resistance to BRAFis limits the therapeutic benefits of targeted therapies [8]. In most cases, resistance to BRAFis involves reactivation of the MAPK cascade, but, in 11% of cases, other signaling pathways such as PI3K/AKT are affected [9]. Activation of PI3K/AKT under BRAFi therapy may also indirectly promote the development of resistance by contributing to the survival of BRAF-inhibited cell populations that are under selection pressure to reactivate the MAPK signaling pathway. After reactivation of the cascade, this is no longer dependent on initial PI3K/AKT activation. The possibility of the simultaneous inhibition of BRAF and signaling pathways that contribute to the selection of BRAFi resistant cells, such as PI3K/AKT, could prevent the development of acquired resistance [10].

The protein ecto-NOX disulfide-thiol exchanger 2 (ENOX2/tNOX) is a member of the NAD(P)H oxidase family and is associated with cell proliferation [11,12,13,14,15] and induction of epithelial–mesenchymal transition, as well as cell migration [16,17,18,19] and invasiveness [20]. In its function as a terminal oxidase of plasma membrane electron transport on the external side of the cell membrane, it primarily uses hydroquinones like ubiquinol as a naturally occurring substrate [21]. Electrons are transferred from cytosolic NAD(P)H via a quinone reductase localized at the internal plasma membrane to hydroquinones, which then serve to transport the electrons through the lipid bilayer [22]. This allows ENOX2 to transfer the electrons to acceptors such as molecular oxygen [23,24] or protein disulfides outside the cell [25].

In addition to this externally localized ENOX2 isoform [11], there is evidence of intracellular localization of the protein [26]. It is not known whether this cytosolic ENOX2 variant only serves as a precursor of the membrane-bound form or fulfills an independent function. Various splice variants of the *ENOX2* mRNA could be responsible for the different localization. Experimentally, the full-length mRNA leads to an internally localized ENOX2 variant, while the *ENOX2*ΔExon4 splice variant results in extracellular localization [27].

The ENOX2 expression of normal cells is significantly lower than that of tumor cells [20,28]. Some authors assume that the occurrence of ENOX2 protein is limited to malignant tumors, as ENOX2, according to them, does not occur in the serum of healthy subjects and patients with other diseases [29,30,31,32]. There are even studies on tumor-specific ENOX2 isoforms; for example, specific ENOX2 isoforms with molecular weights between 37 and 41 kDa were found in the sera of melanoma patients [31,32,33]. In contrast, ENOX2 has been detected immunohistochemically in benign cells [34], and the protein can also be found occasionally in the serum of healthy subjects [35].

Various drugs, including phenoxodiol (PXD) [36,37,38,39], capsaicin [40,41,42,43,44], ME-143 [45] or mitocans [46,47,48], act by inhibiting or downregulating ENOX2. PXD, also known as idronoxil, belongs to the group of synthetic isoflavones [36]. The amino acid sequence of the pentapeptide E394-E-M-T-E acts as a drug-binding site for PXD [49,50]. The resulting inhibition/downregulation of ENOX2 causes an accumulation of ubiquinol and, consecutively, NAD(P)H [51,52]. These changes in turn lead, among others, to an inhibition of the SIRT-1 signaling pathway [41,44,53,54] and sphingosine kinase with a reduced production of sphingosine-1-phosphate (S1P) [39], as well as increased ceramide formation through the activation of sphingomyelinase [51,52]. The resulting protein phosphatase-1-driven dephosphorylation [55] and SIRT-1-inhibition-driven acetylation of AKT [56] lead to a lower activity of AKT [41,51,52]. Subsequent steps ultimately induce a cell cycle arrest and apoptosis [13,14,16,20,37,39,44,51,52,53,54,57,58,59] or autophagy [41].

The ENOX2 protein has an influence on the invasion of immune cells into cancer tissue. In nasopharyngeal carcinoma, high ENOX2 expression is associated with lower immune infiltration and poorer progression-free survival. A combination therapy of PXD and cisplatin increases the infiltration of CD8^+^ effector memory T cells into the tumor area in nasopharyngeal carcinoma, which preferentially lyse cells with higher ENOX2 expression [60].

This work focuses on the role of ENOX2 as a potential prognostic marker and in the treatment of malignant melanoma. Furthermore, the therapy of BRAF-mutated melanoma with the ENOX2 inhibitor PXD for preventing the development of resistance to targeted therapies is investigated. We discovered that high ENOX2 expression in primary melanoma (PM) was associated with decreased overall (OS), disease-specific (DSS) and metastasis-free survival (MFS), suggesting suitability as a complementary biomarker. Low eTIL scores have been associated with high ENOX2 expression. In addition, the inhibition of ENOX2 offers a promising therapeutic option in combination with targeted therapies by reducing BRAFi-induced AKT activation and thus preventing the survival of cell populations that may develop BRAFi resistance.

## 2. Results

### 2.1. Full-Length ENOX2 Is Expressed in Melanoma Cells as Well as in Benign Cells

To investigate the expression of ENOX2 in different tissues, immunohistochemical staining of skin samples was made. Furthermore, Western blot experiments were performed to analyze the expression of the various ENOX2 isoforms.

We were able to detect immunohistochemical ENOX2 expression in benign cells such as keratinocytes and melanocytes (see Figure A1). The staining in all samples indicated an intracellular localization of the ENOX2 protein (see Figure A1 and Figure A2).

The Western blot of ENOX2 in benign cell lines confirmed that ENOX2 expression is not limited to malignant cells (see Figure A3).

Melanoma cell lines showed several bands, possibly isoforms of ENOX2, with varying molecular weight, which are presented in Figure 1. Only the band at 72 kDa was consistently observed in all cell lines with varying intensity. These results indicate expression of the full-length ENOX2 protein.

### 2.2. ENOX2 Protein Expression Is a Potential Prognostic Marker

A tissue microarray with determination of immunohistochemical ENOX2 staining intensity was performed to assess the suitability of ENOX2 as a prognostic marker.

Table 1 provides an overview of the results of the tissue microarray. Of 305 specimens, 21 were excluded due to absence/an insufficient tumor area. The remaining 284 samples contained 249 PMs for which further data were available for analysis. The expression of ENOX2 was significantly different within the categories of sample entity (*p* = 0.024), stage of PM (*p* < 0.001), tumor thickness of PM (*p* < 0.001) and electronic tumor-infiltrating lymphocytes (eTILs; *p* = 0.031). Figure 2 shows the results of these significant categories in detail. An increase in ENOX2 expression was detected in the course of malignant transformation from nevi via PMs to melanoma metastases (MMs), as well as for an increasing tumor thickness and stage. When evaluating the level of immune infiltration of the tumor (eTILs) [62], a significant difference in ENOX2 expression was found across all entities, with low eTIL scores being associated with high ENOX2 expression. Across the histological subtypes, the difference was almost significant (*p* = 0.052), with nodular melanomas in particular showing high ENOX2 expression.

Kaplan–Meier analysis (see Figure 3) revealed a higher OS with lower ENOX2 expression for all PMs and the subgroup of stage I and II melanomas. The Gehan–Breslow–Wilcoxon (GBW) test was able to detect a significant difference in OS (p_GBW_ = 0.0396) for all PMs, while the log-rank (Mantel–Cox; LR) test showed no significant difference in OS. Examining only stage I and II melanomas, no significance was found. The 10-year OS rate was 67.81% (all PMs) and 72.51% (stage I/II) in the high ENOX2 expression group and 80.46% (all PMs) and 83.66% (stage I/II) in the low ENOX2 expression group. Furthermore, DSS was significantly better with low ENOX2 expression in both tests (p_GBW_ = 0.0345; p_LR_ = 0.0482) when all PMs were evaluated. Considering only stage I and II melanomas, a significant result was barely missed (p_GBW_ = 0.0553; p_LR_ = 0.0551). DSS after 10 years was 78.15% (all PMs) or 82.31% (stage I/II) with high ENOX2 expression and 88.95% (all PMs) or 92.98% (stage I/II) with low ENOX2 expression. The biggest difference between high and low ENOX2 expression was seen in MFS. High levels of ENOX2 protein were associated with a significantly increased incidence of metastases, both when considering all PMs (p_GBW_ = 0.0191; p_LR_ = 0.0182) and stage I and II melanomas (p_GBW_ = 0.0015; p_LR_ = 0.0019). For the occurrence of metastases, the hazard ratio (HR) was 3.757 (95% confidence interval (CI_95_) 1.253–11.27) and 8.487 (CI_95_ 2.205–32.67) when considering all stages and stages I and II, respectively. After 10 years, 90.44% (all PMs) and 93.25% (stage I/II) of patients with low ENOX2 were still metastasis-free, but only 76.42% (all PMs) and 75.83% (stage I/II) of patients with high immunohistochemical ENOX2 expression were.

### 2.3. ENOX2 RNA Expression Is a Potential Prognostic Marker

To confirm the suitability of *ENOX2* as a potential prognostic marker at RNA level in melanoma, The Cancer Genome Atlas (TCGA) [64] data were analyzed.

Survival data stratified by TCGA *ENOX2* RNA expression are shown in Figure 4. As with high immunohistochemical ENOX2 expression, OS was decreased for subjects with high *ENOX2* RNA expression. The survival differences were significant for all PMs (p_GBW_ = 0.0152; p_LR_ = 0.0088) as well as for stages I and II (p_GBW_ = 0.0082; p_LR_ = 0.0062). The 2-year OS rate for PMs with high *ENOX2* RNA expression was significantly lower (37.89%) than for those with low *ENOX2* RNA expression (72.69%). The same was the case for stage I and II melanomas (36.00% versus 78.27%). Considering all PMs, the median OS in the high expression group was 23.98 months compared to 35.15 months for low *ENOX2* expression. In stages I and II, the high expression group even had a median OS of only 19.97 months, while it could not be determined for low *ENOX2* expression due to insufficient events. At 3.313 (CI_95_ 1.353–8.117), the HR of all PMs was lower than the HR of stage I and II melanomas at 9.707 (CI_95_ 1.908–49.39).

The analysis of differential expression showed increased expression of *BRAF*, *POU3F2*, *hnRNP F*, *AKT*, *CFLAR* (c-FLIP), *XIAP* and *SIRT1*, among others, in the group with high *ENOX2* expression (see Figure A4).

### 2.4. Growth-Inhibitory Effects of Phenoxodiol on Melanoma Cells

MUH viability assays and colony formation assays were performed to study the effects of ENOX2 and BRAF inhibition on melanoma cell lines in vitro. The ENOX2 inhibitor PXD and the BRAFi vemurafenib were used for this purpose.

Overall, only the SK-MEL-19 and SK-MEL-28 cell lines achieved 50% inhibition (IC_50_) at doses of PXD up to 20 μM. The IC_50_ values were 10.98 μM (SK-MEL-28 S), 14.80 μM (SK-MEL-28 R), 11.73 μM (SK-MEL-19 S) and 16.72 μM (SK-MEL-19 R), respectively. An inhibition of 20% was achieved in all cell lines. The IC_20_ values ranged from 4.35 μM for Mel1617 R to 15.98 μM for 451Lu R for BRAF^V600^-mutated cell lines (see Figure 5).

There was no significant difference between the BRAFi-resistant and -sensitive variants with regard to their IC_20_ values (*p* = 0.9965). The BRAF wildtype cell lines could only be inhibited with significantly higher doses of PXD compared to BRAFi-sensitive (*p* = 0.0426) and-resistant (*p* = 0.0468) variants. A 20% inhibition was achieved at 18.94 μM for MeWo and 19.60 μM for SK-MEL-23.

Figure 6a and Figure A5 show the viability under combined treatment with vemurafenib and PXD. The control treatment with DMSO did not lead to any major reduction in vitality. Not only did the BRAF wildtype cell lines such as MeWo and SK-MEL-23 or cell lines with acquired BRAF resistance display a reduced viability due to the administration of PXD, but also the BRAFi-sensitive variants were inhibited by PXD in addition to the effects of vemurafenib, even if no synergistic effect of the combination therapy could be detected. Resensitization of BRAFi-resistant cells to vemurafenib could not be observed with the combination, as the inhibitory effects at high doses of vemurafenib also occurred without the administration of PXD and were therefore non-specific.

The colony formation assay (see Figure 6b and Figure A6) confirmed the ability of PXD to inhibit melanoma cells. In particular, BRAF-mutated cell lines, both BRAFi-sensitive and -resistant, were suppressed in their colony formation by ENOX2 inhibition. 451Lu S displayed the additive effect of PXD in combination with vemurafenib. In contrast, colony formation was increased in many BRAFi-resistant variants in combined treatment compared to monotherapy with PXD.

In summary, the results confirmed the possibility to inhibit the proliferation of melanoma cells by PXD, both in monotherapy and in combination with targeted therapies.

### 2.5. ENOX2 Inhibition Decreases Phospho-AKT Induction Under BRAF Inhibitor (BRAFi) Therapy

To investigate the molecular mechanisms of vemurafenib and PXD therapy in melanoma cell lines, a Western blot (see Figure 7 and Figure A7) of ENOX2, as well as AKT and ERK, key enzymes of the PI3K/AKT and MAPK cascade, was performed. The focus of the analysis was on the consistently present 72 kDa ENOX2 isoform.

Vemurafenib led to a downregulation of ENOX2 or unchanged expression in most BRAFi-sensitive cells. Only in Mel1617 was a slight upregulation observed. In contrast, ENOX2 was discretely increased in the resistant cells under vemurafenib. PXD did not significantly alter ENOX2 expression.

Vemurafenib partly caused an upregulation of phospho-AKT in BRAFi-sensitive cell lines. However, it had little effect on phospho-AKT in BRAFi-resistant cells and only occasionally generated a slight increase in phosphorylation. In most of the BRAFi-sensitive variants, PXD was able to attenuate the upregulation of phospho-AKT by vemurafenib. In the resistant cells, this effect was not consistently observed; in contrast, there was even an increase in the phosphorylation of AKT under PXD.

Considering ERK, vemurafenib led to a paradoxical phosphorylation in many cell lines, especially the BRAFi-resistant ones. PXD also partially caused a phosphorylation of ERK in the resistant cells.

In summary, the results were quite variable, which made it challenging to draw conclusive findings. However, the reduced induction of phospho-AKT through BRAFis as a result of ENOX2 inhibition was observed in several BRAFi-sensitive cell lines.

## 3. Discussion

### 3.1. ENOX2 Isoforms and Subcellular Location

Only in the 72 kDa range were ENOX2 bands consistently detected in all melanoma and benign cell lines. This supports the translation of the full-length *ENOX2* mRNA, which would result in a protein with a molecular weight of 70.1 kDa [50,61]. Minor deviations in molecular weight and the formation of duplicate bands can be explained by a different degree of glycosylation of ENOX2. In addition, a band existed in the 63 kDa region, which could roughly correspond to translation of the *ENOX2*ΔExon4 splice variant from the first start codon (62 kDa). A protein from the *ENOX2*ΔExon5 mRNA would also be in this range at 60.6 kDa. However, the translation of these two splice variants is only described in transfected COS cells [27], which means that the bands could also be post-translational modifications or degradation products of the full-length protein. The other weak bands possibly represent products of other mRNA splice variants, post-translational modifications or degradation products of ENOX2.

The 47 kDa and 43 kDa ENOX2 isoforms, as well as processed 34 kDa ENOX2 described by Tang et al. [27], could not be detected in our study. Similarly, the detection of a melanoma-specific ENOX2 variant with a molecular weight between 37 kDa and 41 kDa [31,32,33] was not possible. It should be noted that these isoforms were isolated from the sera of tumor patients so the sample materials are different. It is possible that only fragments of ENOX2 are released into the serum, so the full-length ENOX2 is found in cells while smaller variants are detectable in the serum. In addition, a different antibody, albeit with a similar immunogen, was used, which could also influence the detection of ENOX2 isoforms.

The results of immunohistochemistry indicate an intracellular expression of ENOX2. These results are supported by data from The Human Protein Atlas, which also show an intracellular protein expression of ENOX2 [26]. Therefore, the exclusive localization of ENOX2 on the external side of the plasma membrane [11,66,67] cannot be confirmed. In this context, the function of the previously mentioned mRNA splice variants in the process of localization of the ENOX2 protein is of interest. According to Tang et al. [20], the splice variant *ENOX2*ΔExon4 leads to an external localization of the 34 kDa protein, while the full-length *ENOX2* mRNA causes intracellular localization. If intracellular localization is confirmed in further experiments, it is unknown what functions the ENOX2 protein has in this context. It is possible that intracellular ENOX2 directly oxidizes NAD(P)H/H^+^ via its NAD(P)H binding site [49,50]. Since there would still be a regulation of the NAD(P)H/H^+^ to NAD(P) ratio, the cellular signalling pathways affected would be similar to those of cell surface ENOX2.

### 3.2. ENOX2 as a Potential Prognostic Marker

ENOX2 is particularly attractive as a biomarker because it can be detected in easily obtainable sample materials such as serum [68] and urine [69]. Moreover, some authors assume tumor specificity [29,30,31,32]. In contrast, after the immunohistochemical staining and Western blotting of ENOX2, we were also able to detect the expression of ENOX2 in benign cells such as keratinocytes, fibroblasts or melanocytes. Based on these data, a tumor-specific occurrence of the ENOX2 protein does not appear to be the case. This is supported by further protein expression data: according to The Human Protein Atlas [34], smooth muscle cells in particular have high ENOX2 expression, endothelial cells and follicular cells express medium–high ENOX2 levels and numerous other cell types, including keratinocytes, have low ENOX2 expression.

Although these findings do not support the use of ENOX2 as a diagnostic biomarker due to its lack of tumor specificity, it could nevertheless be suitable as a prognostic biomarker. A significantly worse OS, DSS and MFS were found for patients with PM and high ENOX2 protein expression. Patients in localized stages I and II also showed a significantly lower MFS with high ENOX2 protein expression.

The reduced survival and increased metastasis with high ENOX2 protein expression can be explained by the involvement of ENOX2 in the processes of proliferation [11,12,13,15] and metastasis [17,18,19]. High ENOX2 expression leads to an induction of epithelial–mesenchymal transition, which causes an increase in the invasiveness and migration of tumor cells [17,18]. In contrast, ENOX2 downregulation has an inhibitory effect on cell migration and invasiveness [19,20].

Even if our data only prove a correlation of high ENOX2 protein expression with poorer survival in melanoma, the aforementioned literature provides sufficient indications of causality in other cell types.

Since there is a correlation between *ENOX2* mRNA expression and cell growth [15], TCGA [64] *ENOX2* mRNA expression could also serve as a prognostic biomarker. In fact, the correlation of high *ENOX2* mRNA expression with poorer survival that we discovered for melanoma is supported by data from Lin et al. [59], who found a similar effect for hepatocellular carcinoma and breast cancer.

Differential expression analysis of the TCGA data showed an increased RNA expression of proteins previously known to be positively regulated by ENOX2, such as AKT [41,51,52], CFLAR (c-FLIP), XIAP [70] and SIRT1 [41,44,53,54] in the *ENOX2*-high group. This confirms the influence of the ENOX2 cascade on the aforementioned proteins. Furthermore, the *ENOX2* transcription factor *POU3F2* [40,71] and splicing factor *hnRNP F* [72] were increasingly expressed, which may be one of the reasons for the high *ENOX2* expression in this group. Interestingly, the occurrence of *BRAF* was also elevated in the group with high *ENOX2* expression. One possible explanation is that BRAF can influence the expression of ENOX2 via the upregulation of POU3F2 [73].

The effects of ENOX2 on proliferation and invasiveness also explain the association of ENOX2 expression with tumor thickness, tumor stage and sample entity observed in the tissue microarray. ENOX2 expression increased in line with malignant progression from BNs via PMs to MMs. This is supported by data from Kluger et al. [74], who discovered a similar increase in the protein XIAP, which is regulated by ENOX2 via several intermediate steps [70]. Considering this, immunohistochemical ENOX2 intensity cannot serve as an independent biomarker, as its function influences other prognostic parameters such as tumor thickness and stage.

However, high ENOX2 expression could still be used as a dependent prognostic biomarker to identify particularly aggressive melanomas. This is particularly interesting in the context of sentinel lymph node biopsy, as additional biomarkers could be helpful for determining its indication. Patients with a high ENOX2 expression could also be offered intensified treatment options. This includes (neo)adjuvant therapies in early stages [75] and combined immune checkpoint inhibitor therapies in advanced stages [76].

With regard to the clinical application of ENOX2 immunohistochemistry, it should be noted that the method used in this study has some limitations. On the one hand, there is a high inter-rater variability in the evaluation of staining intensities. We overcame this problem by using the median value after evaluation by three independent examiners. On the other hand, the occasional staining of benign tissue requires visual identification of the tumor area in order to only determine the staining intensity in this area. Some of these challenges could be overcome in the future by an automated, algorithm-based evaluation of the staining intensities. If the suitability of *ENOX2* mRNA expression as a biomarker is confirmed, this method could be advantageous compared to immunohistochemical risk stratification with regard to the aforementioned aspects.

It should also be noted that a single biomarker is rarely sufficient to stratify melanoma patients into specific risk groups. Rather, a multimodal approach is necessary, for example, as part of a gene expression analysis score, to which *ENOX2* can contribute as a biomarker.

A further limitation of this study concerns the low number of stage IV melanomas, which biases conclusions about this group. Nevertheless, the effects on metastasis-free survival were also present in the subgroup of stage I and II patients, for whom prognostic statements are of greatest importance.

### 3.3. ENOX2 Inhibition in Combination with Targeted Therapies to Prevent Acquired Resistance

All melanoma cell lines tested had an IC_50_ of PXD above 10 μM. Many cell lines even exceeded the maximum tested dose of 20 μM with their IC_50_ values. Higher doses of PXD were not used in our study to prevent non-ENOX2-mediated effects of PXD, such as the inhibition of topoisomerase [36]. Therefore, the IC_20_ values were used to compare the effects of PXD on the cell lines. For comparison, the IC_50_ of PXD is between 0.2 μM for HeLa cells (cervical carcinoma) [37] and 62.5 μM for HT-29 cells (colorectal carcinoma) [77]. The melanoma cell line MM200 has an IC_50_ of 3.0 μM PXD [78], making it more susceptible than the cell lines we tested.

The BRAF-mutated cell lines were more sensitive to PXD than the BRAF wildtype cell lines. This is in contrast to the data from Yu et al. [79], which describe a weaker response of BRAF-mutated cell lines to PXD. The authors attribute this to the influence of ERK on the phosphorylation of the Bcl-2 proteins BAD and BIM. Due to the multiple signaling pathways affected by ENOX2 inhibition, the molecular effects of ERK on these Bcl-2 proteins appear to play a subordinate role. In addition, oncogenic BRAF can upregulate the transcription factor POU3F2 [73], which is positively correlated with ENOX2 expression [40,71]. As a consequence, more ENOX2 protein is available in BRAF-mutated cells, and can therefore be inhibited, resulting in greater effects of PXD.

The addition of PXD to vemurafenib had an additive, albeit not synergistic, antiproliferative effect on all cell lines in the viability assay.

The protein expression of ENOX2 was not clearly altered by treatment with 5 μM PXD. A slight downregulation could be detected in some cases. It is possible that the concentrations of PXD were insufficient to lead to an effect at the protein expression level. Other studies used much higher concentrations of up to 40 μM PXD or observed the effects over a longer period of time [55,80,81,82]. Nevertheless, modulation of the activity of ENOX2 does not have to be reflected in a change in its expression, as inhibition of enzymatic activity can also occur without a noticeable change in expression. Furthermore, processes such as phosphorylation by PKCδ also contribute to the regulation of ENOX2 activity [83].

The paradoxical activation of ERK by vemurafenib can be attributed to the loss of negative feedback by BRAF^V600^-activated ERK on receptor tyrosine kinases, RAS and RAF. This restores the physiological MAPK signaling pathway so that ERK can be activated again via receptor tyrosine kinases [84]. This also explains why vemurafenib treatment in resistant cell lines sometimes led to increased colony formation.

The suppression of the PI3K/AKT signaling pathway is also removed by the elimination of negative feedback by vemurafenib [85], resulting in increased AKT phosphorylation. This was particularly evident in the BRAFi-sensitive cell lines. Since the signaling pathways altered by ENOX2 inhibition lead to a reduced activity of AKT [41,51,52], PXD is, in theory, capable of inhibiting the PI3K/AKT signaling pathway.

A combination therapy is particularly interesting because the activation of the PI3K/ AKT signaling pathway by BRAFis can lead to the survival of subpopulations of BRAF-inhibited tumor cells. These are under selection pressure to reactivate the MAPK signaling pathway. If the MAPK cascade is reactivated by new mutations, survival no longer depends on the transient activation of the PI3K/AKT signaling pathway. This means that the signaling pathways that indirectly contribute to the development of resistance, such as PI3K/AKT, must also be taken into account during initial therapy with MAPK inhibitors [10]. Indications of the benefit of PXD with regard to this aspect are shown by the Western blot, where PXD, even at the low concentrations used, was able to partially attenuate the phospho-AKT induction caused by vemurafenib. The combination of PXD and targeted therapies could therefore reduce the occurrence of acquired BRAFi resistance.

However, a resensitization of BRAFi-resistant cell lines to vemurafenib by PXD could not be proven by our study. This is not surprising, as acquired resistance to BRAFis only in 11% of cases is caused by permanent changes outside the MAPK signaling pathway, such as in the PI3K/AKT signaling pathway [9].

Studies suggest that serum levels of PXD above the IC_50_ of melanoma cell lines can be achieved [80]. Although a response of tumors to PXD monotherapy has not yet been proven in clinical studies, since the therapy is associated with few side effects [86,87,88], its use in melanoma therapy is possible.

New drugs such as ME-143 could further improve the effectiveness of ENOX2 inhibition, as it binds to ENOX2 in vitro with four to ten times higher affinity than PXD [45]. This means that lower plasma levels are sufficient to achieve the same degree of inhibition.

Nevertheless, risks of ENOX2 inhibitor therapy remain. The ENOX2 inhibitor capsaicin can lead to a downregulation of ENOX2 at the protein and mRNA level [57,89], but low doses of capsaicin (≤10 μM) can trigger a paradoxical ENOX2 upregulation [41]. Due to this potentially cancer-promoting role of capsaicin, caution is advised in its clinical use [17].

### 3.4. Immunomodulatory Effects of ENOX2

The ENOX2 protein has an influence on the infiltration of immune cells into cancer tissue, with high ENOX2 expression being associated with lower immune infiltration [60]. Our data confirm this association between ENOX2 expression and immune infiltration, as the proportion of high ENOX2 expression was significantly higher in tumor samples with few eTILs (≤16.6%) [62].

One possible mechanism for this is that ENOX2 can influence the S1P level via the enzyme sphingosine kinase [39,51,52]. S1P is secreted into the tumor microenvironment, where it exerts paracrine, which has modulatory effects on immune cells and thus contributes to the immune evasion of the tumor (see Figure 8) [88,90].

This could also explain the increase in immune infiltration due to ENOX2 inhibition with PXD in nasopharyngeal carcinoma [60]. It is possible that an increased number of eTILs as part of an antitumor immune response can also be generated in melanoma through treatment with ENOX2 inhibitors. This is particularly interesting in combination with immune checkpoint inhibitor therapy, as the effects of this therapy could be enhanced by inhibiting the increased ENOX2 expression in cancer tissue.

However, it should be noted that several authors report the inhibition of immune cells by PXD [91,92,93]. The possibility of topical application (e.g., transcutaneously in melanoma or intravesically in urothelial carcinoma) could minimize the generalized inhibitory effects on immune cells. Nevertheless, the immunosuppressive S1P production would be restricted in the tumor microenvironment and the proliferation of tumor cells would be reduced by the direct effects of ENOX2 inhibition.

A low eTIL value is not only a negative prognostic parameter [94] but can also serve at very low values ≤12.2% as a negative predictive biomarker for the response to PD-1 immune checkpoint inhibitor therapy in previously untreated metastases [95]. If a correlation between eTILs and ENOX2 expression is confirmed, ENOX2 expression could also act as a predictive biomarker for PD-1 therapy.

## 4. Materials and Methods

### 4.1. Immunohistochemistry of the Tissue Microarray

The tissue microarray was provided by Dr. Kristian Ikenberg and Dr. Christian Busch as described elsewhere [96]. After dewaxing, the tissue microarray was pretreated by boiling for 20 min in a pressure cooker with HIER citrate buffer (10×) pH 6.0 (BIOZOL, Eching, Germany) diluted 1:10 with ddH_2_O. Immunohistochemistry with the ENOX2 antibody (Proteintech Group, Rosemont, IL, USA, #10423-1-AP) diluted 1:100 in 1:20 donkey serum (Merck, Darmstadt, Germany)/PBS-Triton X-100 (0.1%, Merck, Darmstadt, Germany) incubating for 2.5 h was performed using the Epredia™ Lab Vision™ UltraVision™ LP detection system: AP Polymer (Ready-To-Use) (Thermo Fisher, Waltham, MA, USA) according to the manufacturer’s protocol. Permanent AP-Red-Kit (BIOZOL, Eching, Germany) was applied for 10 min under microscopic control and the tissue microarray was then counterstained with hematoxylin. Digitization was performed with the VENTANA DP 200 slide scanner (Roche, Mannheim, Germany) and the cores were analyzed with the program QuPath-0.3.2 [97]. In the tissue samples, only the ENOX2 staining intensity of the melanoma cells/melanocytes, which were visually identified by the evaluator, was analyzed. The intensity was graded with one point for negative or weak staining, two points for moderate staining and three points for strong staining (see Figure A2). This was determined for the maximum and most frequent ENOX2 staining intensity of the sample and the points were added to a total score between 2 and 6. The scores were determined by three independent evaluators and the median of the values was calculated. Samples that achieved a total score of 2 to 4 were summarized as the low ENOX2 expression group, while total scores of 5 or 6 belonged to the high ENOX2 expression group. The follow up of the patients was between 1 and 186 months, with a mean value of 84.07 months. Patients were censored after a maximum period of 120 months. eTILs were determined by Chatziioannou et al. [95].

### 4.2. The Cancer Genome Atlas Data Analysis

Data of *ENOX2* RNA expression in malignant melanoma were obtained via The Human Protein Atlas [65] from the Center for Cancer Genomics—National Cancer Institute [64]. The data were sorted by expression level (fragments per kilobase of transcript per million reads mapped). The upper quartile was defined as the group with high *ENOX2* expression, the lower 75% of the samples as low *ENOX2* expression.

The differential gene expression of the two groups was analyzed using the R2: Genomics Analysis and Visualisation Platform [98]. The default values for creating a volcano plot were used, with the minimum number of present calls set to 20 and the minimum maximum value set to 2.

### 4.3. Inhibitors and Culture of Melanoma Cell Lines

Vemurafenib was acquired from Selleck Chemicals (Houston, TX, USA) and PXD from MedChemExpress (Monmouth Junction, NJ, USA). The origin of the melanoma cell lines 451Lu, A-375, Mel1617, SK-MEL-19 and SK-MEL-28 and basic cell culture methods have been described previously [99]. MeWo was provided by Prof. Ralf Gutzmer and SK-MEL-23 was purchased from ATCC (Manassas, VA, USA). BRAF wildtype and BRAFi-sensitive (S) cells were cultured at 37 °C and 5% CO_2_ in T-75 cell culture flasks (Sarstedt, Nuembrecht, Germany) containing 12 mL RPMI 1640 culture medium (Thermo Fisher, Waltham, MA, USA) with the addition of 1% (*v*/*v*) penicillin/streptomycin (Thermo Fisher, Waltham, MA, USA) and 10% (*v*/*v*) fetal calf serum (Merck, Darmstadt, Germany). Medium of BRAFi-resistant (R) cells additionally contained vemurafenib 2 μM. For splitting and harvesting, the cells were rinsed with PBS (Merck, Darmstadt, Germany) and then suspended by adding 4 mL of 0.05% trypsin/EDTA (Thermo Fisher, Waltham, MA, USA). After addition of 6 mL culture medium, the cells were centrifuged for 4 min at 300× *g* and the supernatant discarded.

### 4.4. Viability Assay

A total of 2.5 × 10^3^ cells in 100 μL medium were added to 60 wells of 96-well plates (Sarstedt, Nuembrecht, Germany) and cultivated for 24 h. The treatment was then carried out in quadruplicates for 72 h with increasing concentrations up to 20 μM vemurafenib and up to 20 μM PXD as well as combinations of the drugs in all concentrations and a DMSO (AppliChem, Darmstadt, Germany) control. After washing with PBS, addition of 4-methylumbelliferyl heptanoate (100 μg/mL diluted in PBS, Merck, Darmstadt, Germany) and incubation for 45 min at 37 °C, the fluorescence (λ_ex_ 355 nm, λ_em_ 460 nm) was measured using the TriStar LB 941 fluorescence microplate reader (Berthold Technologies, Bad Wildbad, Germany). The amount of fluorescence is indicative of the number of viable cells after treatment and was normalized to the untreated control.

### 4.5. Colony Formation Assay

A total of 5 × 10^2^ cells in 1 mL medium were added to the wells of 12-well plates (Thermo Fisher, Waltham, MA, USA). After 24 h of culture, the medium was replaced by triplicates of medium containing 2 μM vemurafenib, 5 μM PXD, a combination of the former or an equivalent volume of DMSO as control. During the incubation period of 7 days, the treatment was renewed after 3–4 days. The colonies were then washed with PBS, fixed with 4% formalin (Merck, Darmstadt, Germany) for 15 min and stained with a solution consisting of 3% (*v*/*v*) crystal violet (Merck, Darmstadt, Germany), 50% (*v*/*v*) methanol (VWR, Radnor, PA, USA) and water for 1 h. After washing with demineralized water and drying, the plates were scanned with the Epson Perfection V850 Pro scanner (Düsseldorf, Germany). A visual identification of colonies was performed by an evaluator. Plates were only included in the evaluation in the presence of colony formation under the control treatment.

### 4.6. Western Blot

A total of 4 × 10^5^ cells in 2 mL culture medium were seeded into each well of 6-well plates (Sarstedt, Nuembrecht, Germany). After 24 h of culture, the medium was replaced with a medium containing the treatments mentioned above for the colony formation assay. Following another 24 h of treatment, the cells were harvested and cleaned with PBS. The cells were lysed for 30 min on ice with RIPA buffer consisting of 25 mM Tris-HCl (Merck, Darmstadt, Germany) pH 7.6, 150 mM sodium chloride (Merck, Darmstadt, Germany), 1% (*v*/*v*) NP-40 (Merck, Darmstadt, Germany), 1% (*w*/*v*) sodium deoxycholate (AppliChem, Darmstadt, Germany), 0.1% (*w*/*v*) SDS (Roth, Karlsruhe, Germany), 20 mM sodium fluoride (Merck, Darmstadt, Germany), 1 mM orthovanadate (Merck, Darmstadt, Germany), 1 mM pyrophosphate (Merck, Darmstadt, Germany), 0.1 mM phenylmethylsulphonyl fluoride (Roche, Basel, Switzerland), 10 μM pepstatin A (Roth, Karlsruhe, Germany), 10 μM leupeptin (Merck, Darmstadt, Germany) and 3.85 μM aprotinin (Merck, Darmstadt, Germany). The lysate was then centrifuged at 18,000× *g* for 10 min at 4 °C and the protein concentration in the supernatant was determined. Lysates of primary human keratinocytes, fibroblasts and melanocytes were kindly provided by Prof. Birgit Schittek and Dr. Jule Focken, as described elsewhere [100]. A total of 30 μg of protein was subjected to SDS-PAGE and transferred to polyvinylidene difluoride (PVDF) membranes (Bio-Rad, Hercules, CA, USA). Proteins were detected with primary antibodies against ENOX2 (Proteintech Group, Rosemont, IL, USA, #10423-1-AP, diluted 1:1000), Cell Signaling Technology (Danvers, MA, USA) antibodies against phospho-AKT (Ser473, #4060, diluted 1:2000), AKT (#9272, diluted 1:1000), phospho-ERK (Thr202/Tyr204, #4376, diluted 1:1000), ERK (#9102, diluted 1:1000) and β-actin (#4967, #3700, diluted 1:1000), as well as secondary antibodies coupled to horseradish peroxidase (Cell Signaling Technology, Danvers, MA, USA, diluted 1:1000) with Pierce ECL Western Blotting and SuperSignal West Dura Extended Duration Substrate (Thermo Fisher, Waltham, MA, USA) using the Amersham Imager 600 (GE Healthcare, Chicago, IL, USA).

### 4.7. Statistical Analysis

Statistical analysis was performed using the programs GraphPad Prism 8.4.0 (Boston, MA, USA) and IBM SPSS Statistics 29.0.0.0 (Armonk, NY, USA). IC values and colony formation assay were analyzed by one-way analysis of variance. The results of the tissue microarray were tested for significance using the Fisher exact test for dichotomous variables and the Fisher–Freeman–Halton exact test for polytomous variables. The differences in the Kaplan–Meier curves were analyzed using the LR test and the GBW test. *p*-values < 0.05 were considered statistically significant (*: *p* < 0.05; **: *p* ≤ 0.01; ***: *p* ≤ 0.001, ****: *p* ≤ 0.0001).

## 5. Conclusions

This study focused on the role of ENOX2 as a target for the prognosis and therapy of malignant melanoma. The cell lines mainly expressed a 72 kDa ENOX2 variant, which we interpreted as full-length ENOX2 protein. The subcellular localization was mostly cytoplasmic, which complements the previously known membranous localization. The expression of ENOX2 mRNA and protein in melanoma provided a potential prognostic marker. High ENOX2 expression was associated with significantly reduced OS, DSS and MFS, as well as with a reduction in eTILs. Clinical application is particularly interesting with regard to the indication for sentinel lymph node biopsy and the determination of therapy intensity and could be combined with other biomarkers. The ENOX2 protein could also be suitable as a therapeutic target. Melanoma cells were inhibited in their proliferation by treatment with the ENOX2 inhibitor PXD, resulting in additive effects to the established BRAFi therapy. ENOX2 inhibition could also help to prevent the development of BRAFi resistance through the inhibition of phospho-AKT induction. Additional effects of PXD that possibly reduce tumor immune evasion may be beneficial for immune checkpoint inhibitor therapy.

In conclusion, ENOX2 is promising for the prognosis and treatment of malignant melanoma. Further experiments are needed to confirm our results and enable the clinical use of ENOX2 in these areas.

## Figures and Tables

**Figure 1 ijms-25-11853-f001:**
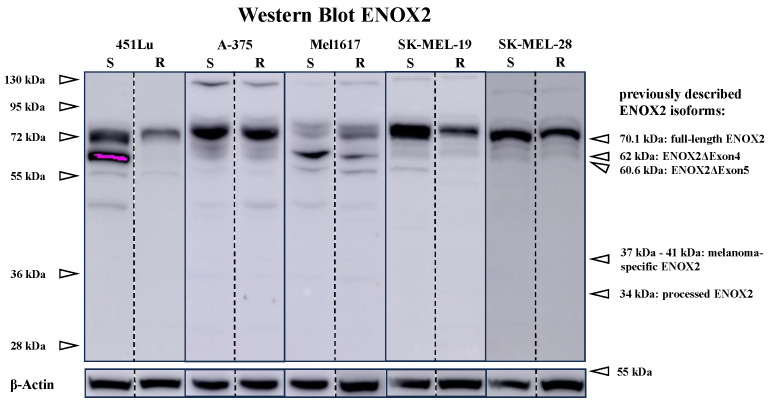
Western blot of melanoma cell lines with ENOX2 antibody shows a constant band at approximately 72 kDa. Further bands were present between 120 kDa and 130 kDa and at 63 kDa, 57 kDa and 50 kDa. The results indicate expression of the full-length ENOX2 protein [50,61]. Other reported ENOX2 variants are the ENOX2ΔExon4 and 5 variants, processed ENOX2 [27] and melanoma-specific ENOX2 variants [31,32,33]. The experimental samples from each cell line (451Lu, A-375, Mel1617, SK-MEL-19, SK-MEL-28) were run on the same blot/gel as indicated by the dashed line. S = sensitive to vemurafenib, R = resistant to vemurafenib.

**Figure 2 ijms-25-11853-f002:**
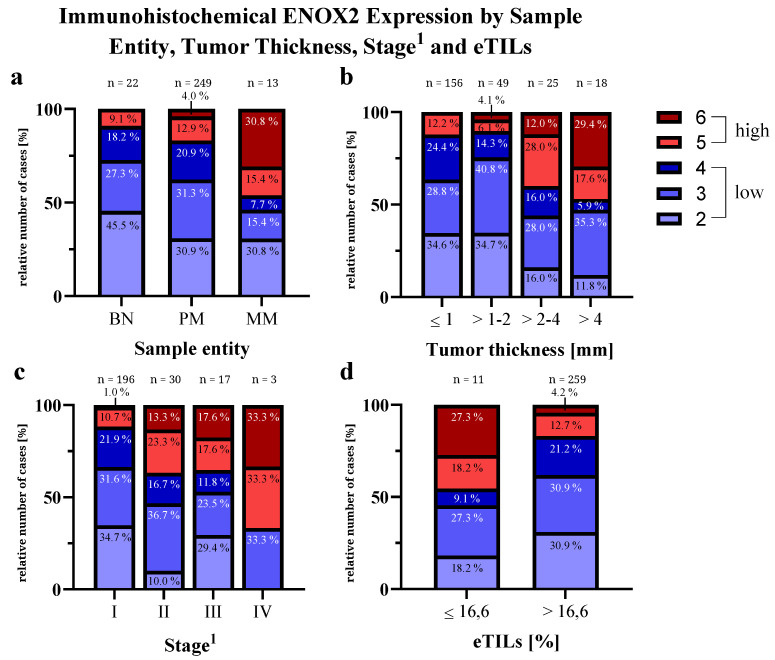
Immunohistochemical ENOX2 expression within the significantly differing categories (see Table 1): (**a**) sample entity, (**b**) tumor thickness, (**c**) stage and (**d**) eTILs [62]. BN = benign nevus, PM = primary melanoma, MM = melanoma metastasis, eTILs = electronic tumor-infiltrating lymphocytes. ^1^ Classification according to AJCC cancer staging manual, 7th edition [63].

**Figure 3 ijms-25-11853-f003:**
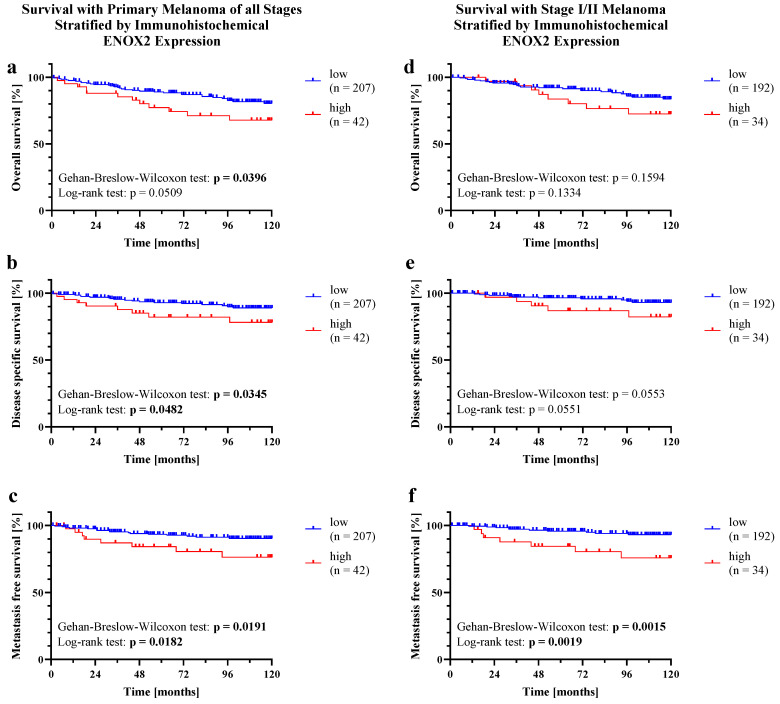
Survival with PM stratified by immunohistochemical ENOX2 expression. Stratification by staining intensity of PMs of (**a**–**c**) all stages and (**d**–**f**) stages I and II. (**a**,**d**) OS, (**b**,**e**) DSS and (**c**,**f**) MFS are shown, respectively. Censoring of patients with survival over 120 months. The differences in survival were tested for significance using the log-rank (Mantel–Cox; LR) test and Gehan–Breslow–Wilcoxon (GBW) test. Values with a *p* < 0.05 were considered significant.

**Figure 4 ijms-25-11853-f004:**
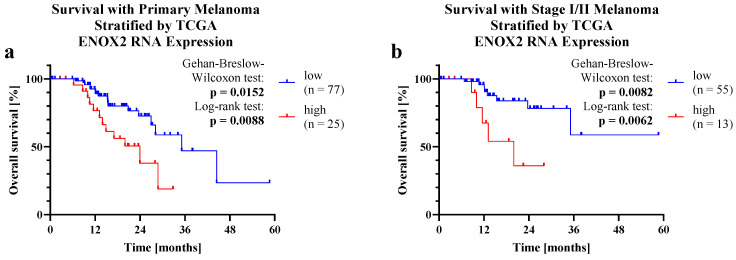
Survival with PM stratified by The Cancer Genome Atlas (TCGA) *ENOX2* RNA expression. Separation into PMs of (**a**) all stages (I (*n* = 2), II (*n* = 65), I/II (*n* = 1), III (*n* = 27), IV (*n* = 3) and stage not available (*n* = 4)), as well as (**b**) stages I and II. The data were sorted by expression level. The upper quartile was defined as the group with high *ENOX2* expression, the lower 75% of the samples as low *ENOX2* expression. The differences in survival were tested for significance using the LR test and GBW test. Values with a *p* < 0.05 were considered significant. Data obtained from the Center for Cancer Genomics—National Cancer Institute [64] via The Human Protein Atlas [65]. TCGA = The Cancer Genome Atlas.

**Figure 5 ijms-25-11853-f005:**
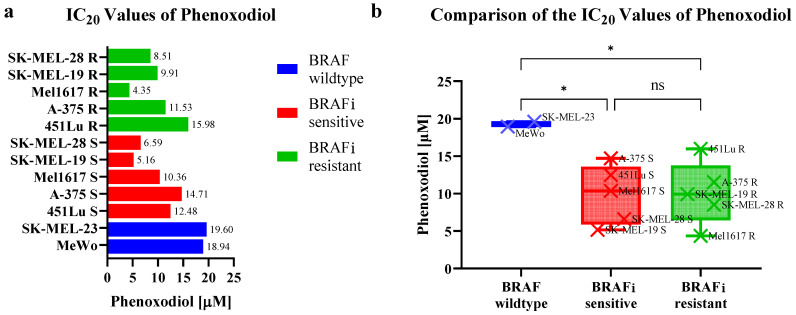
(**a**) Twenty percent inhibiting concentration (IC_20_) values of the ENOX2 inhibitor phenoxodiol (PXD) of melanoma cell lines stratified by BRAF mutation status and BRAF inhibitor (BRAFi) resistance. IC_50_ was only reached for the SK-MEL-19 S/R and SK-MEL-28 S/R cell lines. (**b**) BRAF wildtype cells showed higher IC_20_ values of PXD than BRAF-mutated variants. One-way analysis of variance was used to analyze the differences in IC_20_. ns: *p* ≥ 0.05, *: *p* < 0.05.

**Figure 6 ijms-25-11853-f006:**
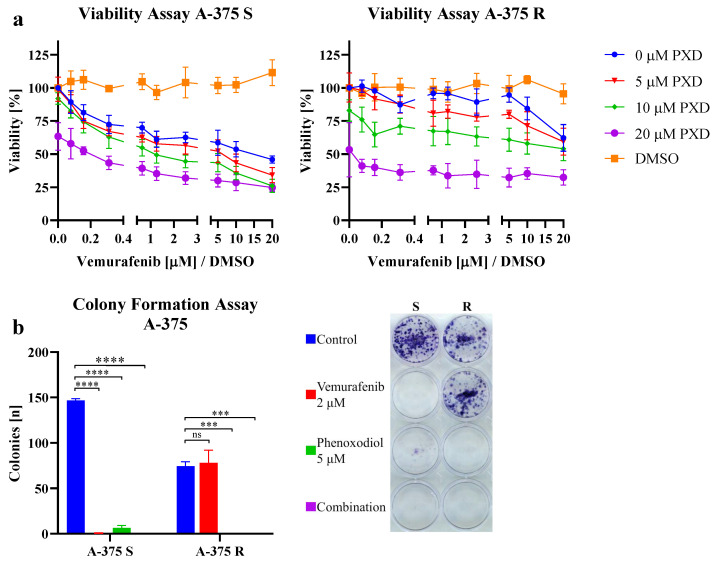
(**a**) Exemplary viability assay of BRAFi-sensitive (S) and -resistant (R) melanoma cell line A-375 under combination therapy with phenoxodiol (PXD) and vemurafenib or DMSO control treatment. An additive effect of PXD treatment can be seen in BRAFi-sensitive cells. BRAFi-resistant cells respond to PXD, but not to BRAFi, except at very high doses. *n* = 4 replicates. Points indicate the mean, error bars indicate the standard error of the mean. (**b**) Exemplary colony formation assay of the same cell line under therapy with 2 μM vemurafenib, 5 μM PXD, combination of both or control treatment. One-way analysis of variance was used to analyze the differences in the number of colonies formed under treatment compared to the control. *n* = 3 technical replicates. Bars indicate the mean, error bars indicate the standard error of the mean. ns: *p* ≥ 0.05, ***: *p* < 0.001, ****: *p* < 0.0001. Additional results are presented in Figure A5 and Figure A6.

**Figure 7 ijms-25-11853-f007:**
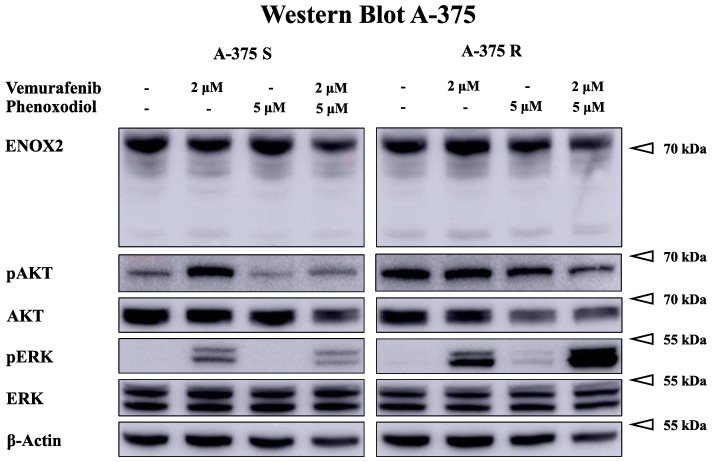
Exemplary Western blot of BRAFi-sensitive (S) and -resistant (R) melanoma cell line A-375 under therapy with 2 μM vemurafenib, 5 μM PXD, combination of both or control treatment. The sensitive cell line shows an upregulation of phospho-AKT under BRAFis, which can be attenuated by administration of PXD. The experimental samples using ENOX2 and β-Actin, pAKT and ERK, as well as pERK and AKT, were each run on the same blot/gel. Additional results are presented in Figure A7.

**Figure 8 ijms-25-11853-f008:**
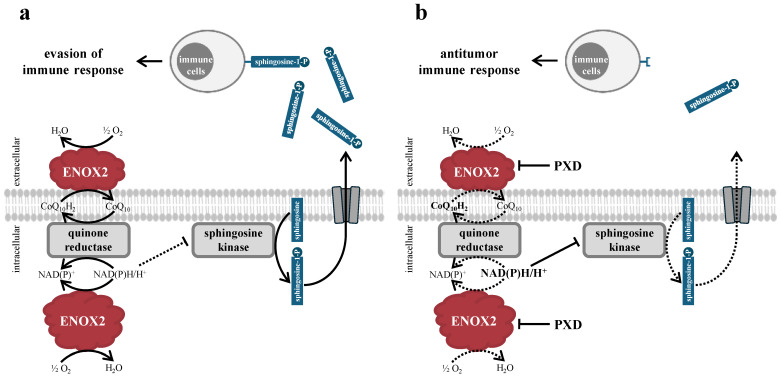
(**a**) Graphical summary of the possible, immunomodulatory effects exerted by ENOX2. (**b**) The inhibition of ENOX2 by PXD leads to a reduced production of sphingosine-1-phosphate (S1P) via the accumulation of NAD(P)H and the subsequent inhibition of sphingosine kinase. S1P otherwise promotes immune evasion of the tumor through paracrine effects on immune cells via G-protein-coupled S1P receptors. Modified according to de Luca et al. [70], Kiknavelidze et al. [88] and Rodriguez et al. [90].

**Table 1 ijms-25-11853-t001:** Evaluation of the tissue microarray. The Fisher exact test was used to calculate the *p*-value for dichotomous variables and the Fisher–Freeman–Halton exact test for polytomous variables. Values with a *p* < 0.05 were considered significant and are highlighted. PM = primary melanoma, MM = melanoma metastasis, eTILs = electronic tumor-infiltrating lymphocytes, OS = overall survival, DSS = disease-specific survival, MFS = metastasis-free survival.

Criteria	ENOX2 Expression [*n*]			*p*-Value
	Low	High	Total	
**samples**				
total			305	
missing			13	
insufficient tumor area	8 (100.0%)	0 (0.0%)	8	
valid	234 (82.4%)	50 (17.6%)	284	
**sample entity**				**0.024**
primary melanoma (PM)	207 (83.1%)	42 (16.9%)	249	
melanoma metastasis	7 (53.8%)	6 (46.2%)	13	
benign nevus	20 (90.9%)	2 (9.1%)	22	
**histology (PM)**				0.052 ^1^
superficial spreading melanoma	167 (85.2%)	29 (14.8%)	196	
nodular melanoma	10 (58.8%)	7 (41.2%)	17	
lentigo maligna melanoma	18 (85.7%)	3 (14.3%)	21	
acral lentiginous melanoma	7 (77.8%)	2 (22.2%)	9	
other subtype	5 (83.3%)	1 (16.7%)	6	
**stage (PM) ^2^**				**<0.001 ^1^**
I	173 (88.3%)	23 (11.7%)	196	
II	19 (63.3%)	11 (36.7%)	30	
III	11 (64.7%)	6 (35.3%)	17	
IV	1 (33.3%)	2 (66.7%)	3	
unknown	3 (100.0%)	0 (0.0%)	3	
**tumor thickness (PM)**				**<0.001 ^1^**
≤1.00 mm	137 (87.8%)	19 (12.2%)	156	
1.01–2.00 mm	44 (89.8%)	5 (10.2%)	49	
2.01–4.00 mm	15 (60.0%)	10 (40.0%)	25	
>4.00 mm	9 (52.9%)	8 (47.1%)	17	
unknown	2 (100.0%)	0 (0.0%)	2	
**ulceration (PM)**				0.147 ^1^
yes	17 (70.8%)	7 (29.2%)	24	
no	188 (84.3%)	35 (15.7%)	223	
unknown	2 (100.0%)	0 (0.0%)	2	
**eTILs**				**0.031 ^1^**
≤16.6%	6 (54.5%)	5 (45.5%)	11	
>16.6%	215 (83.0%)	44 (17.0%)	259	
unknown	13 (92.9%)	1 (7.1%)	14	
**events for OS up to 120 months (PM)**				**0.048**
yes	32 (72.7%)	12 (27.3%)	44	
no	175 (85.4%)	30 (14.6%)	205	
**events for DSS up to 120 months (PM)**				0.055
yes	18 (69.2%)	8 (30.8%)	26	
no	189 (84.8%)	34 (15.2%)	223	
**events for MFS up to 120 months (PM)**				**0.040**
yes	16 (66.7%)	8 (33.3%)	24	
no	191 (84.9%)	34 (15.1%)	225	

^1^ Cases that could not be assigned to a specific group were excluded for the calculation of statistical significance. ^2^ Classification according to AJCC cancer staging manual, 7th edition [63].

## Data Availability

The raw data supporting the conclusions of this article will be made available by the authors on request.

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
