# Peer review of "Ecto-NOX Disulfide-Thiol Exchanger 2 (ENOX2/tNOX) Is a Potential Prognostic Marker in Primary Malignant Melanoma and May Serve as a Therapeutic Target"

_ijms, 2024, doi:10.3390/ijms252111853_

Round 1
Reviewer 1 Report
Comments and Suggestions for Authors
The manuscript titled " ENOX2 (tNOX) is a marker of poor prognosis in primary malignant melanoma and may serve as a therapeutic target" by Böcker et al. investigates the potential of ENOX2 as a prognostic marker and therapeutic target in malignant melanoma, employing a variety of noteworthy methods. The authors utilize differential expression analysis of TCGA data and western blotting in an effort to elucidate the underlying mechanisms of ENOX2’s role. While the topic is both relevant and significant, with potential merit in the study, there are several critical concerns that must be addressed.
In general, the interpretation of the results and the conclusions drawn appear to be overly broad and not fully aligned with the data presented. The title of the manuscript should be revised to reflect a more cautious interpretation, such as “ENOX2 is a Potential Prognostic Marker ....” to better match the scope of the findings.
Other major points:
1. While the data suggest the potential of ENOX2 as a prognostic factor in malignant melanoma, no direct evidence has been provided to demonstrate that ENOX2 functions as a tumor antigen or that its expression levels influence the prognosis of malignant melanoma. A potential improvement would be to include genetic perturbation of ENOX2 using CRISPR/Cas9 or overexpression studies, assessing their effects on proliferation rate or colony formation in melanoma cell lines.
- In the “Results” section, several headings are speculative and do not accurately reflect the data presented. For instance, the title “2.1. ENOX2 Protein Expression Serves as a Prognostic Biomarker” is not fully substantiated by the data. The results only demonstrate an association between ENOX2 expression levels and poor survival outcomes, rather than establishing ENOX2 as a definitive prognostic biomarker.
Additionally, this section lacks sufficient background and context, and includes unnecessary commentary (e.g., “Interestingly, we…”), which should be avoided in objective reporting. Furthermore, there is no reference to any figure supporting the starting statement: “Interestingly, we were able to detect immunohistochemical ENOX2 expression in 92 benign cells such as keratinocytes and melanocytes. The staining in all samples indicates an intracellular localization of the ENOX2 protein.” This claim requires proper visual or analytical support, and it would be preferable to include this data as a Supplementary figure.
Moreover, it is important to comment on the sample sizes presented in Table 1. Specifically, in the tumor “Stage (PM)” category, stage IV includes only three samples: one with low ENOX2 expression and two with high. Drawing a conclusion about ENOX2 expression and tumor progression based on these numbers is not justified.
- In the “Results” section 2.3, titled “Melanoma Cell Lines Mainly Express a 72 kDa ENOX2 Protein”, the title does not accurately reflect the presented findings. It remains unclear whether ENOX2 expression is predominantly observed in melanoma cells. If this is the case, comparisons with other cell types, such as keratinocytes, should be included to substantiate this claim. Alternatively, if the statement refers to comparisons with other proteins, appropriate control experiments are necessary.
The “Western Blot” analysis presented in Figure 4 lacks essential controls: 1) A secondary antibody control, where the primary antibody is omitted, is needed to confirm specific binding of the primary antibody. 2) A loading control should be included by stripping and re-blotting against housekeeping proteins such as actin or tubulin.
Overall, the text in this section lacks clarity and depth, as it primarily describes western blot bands across different cell lines without drawing meaningful conclusions.
- In the “Results” section 2.4, titled “Growth Inhibitory Effects of Phenoxodiol on Melanoma Cells”, providing background information on the treatments used would enhance the reader’s understanding of the results presented. Additionally, each experiment’s objectives should be clarified before listing the values that are already presented in the figures.
Specifically, on page 8, line 172: The reference to “Figure 6a and 2a show…” should be relocated to a new paragraph to improve readability. Furthermore, the text should be enriched with background information to better contextualize the findings.
Comments on the Figures:
Please clarify what Y axis represents in Figure 1.
Please clarify what “S” and “R” represent in Figure 4.
Comments on the methods:
1. The “Tissue Microarray” method is described as immunohistochemistry. If immunohistochemistry was indeed used, it would be ideal to include representative images in the supplementary material or appendix. Additionally, there are issues with the language in this section, where the description starts with the singular "the side" and later shifts to "slides" and "scans," which disrupts clarity.
Furthermore, the statement “only the ENOX2 staining intensity of the melanoma cells/melanocytes was evaluated” requires clarification. Please explain which other cell types were present and how they were excluded during the scanning process.
2. In “Colony Formation Assay”, crystal violet staining stains all cell, including non-colony forming cells. It is unclear how these cells were excluded.
Comments on the Quality of English Language
The manuscript is marked by abundant grammatical errors which significantly impede readability. Enumerating all these errors is beyond the scope of this evaluation. Moderate revision is necessary to enhance the language quality of the manuscript.
Author Response
Reviewer 1
- In general, the interpretation of the results and the conclusions drawn appear to be overly broad and not fully aligned with the data presented. The title of the manuscript should be revised to reflect a more cautious interpretation, such as “ENOX2 is a Potential Prognostic Marker ....” to better match the scope of the findings.
We agree with this comment. Therefore, we have changed the title to “ENOX2 (tNOX) is a Potential Prognostic Marker in Primary Malignant Melanoma and May Serve as a Therapeutic Target”. In addition, the statements were formulated more cautiously to match the scope of the findings (also see 2 and 3).
- While the data suggest the potential of ENOX2 as a prognostic factor in malignant melanoma, no direct evidence has been provided to demonstrate that ENOX2 functions as a tumor antigen or that its expression levels influence the prognosis of malignant melanoma. A potential improvement would be to include genetic perturbation of ENOX2 using CRISPR/Cas9 or overexpression studies, assessing their effects on proliferation rate or colony formation in melanoma cell lines.
We agree with this comment. As already mentioned, the statements regarding a possible causality of worsened survival due to high ENOX2 expression were interpreted more cautiously, so that only a correlation between the parameters is suggested. Functional conclusions regarding ENOX2 in melanoma are beyond the scope of our project. Nevertheless, numerous studies with other cell types/animal models, which also include overexpression and knockdown experiments, provide sufficient evidence of causality in these cell types (Liu et al., 2012; Su et al., 2012; Yagiz et al., 2006, 2007; Chen et al., 2006; Mao et al., 2008; Tang et al., 2007). Further experiments with melanoma cells are necessary to establish ENOX2 as a definitive biomarker in melanoma. This line of argumentation was also adopted in the discussion: “Even if our data only prove a correlation of high ENOX2 expression with poorer survival in melanoma, the aforementioned literature provides sufficient indications of causality in other cell types.” (line 275-277).
We thank you for your suggestions regarding further experiments with genetic perturbation of ENOX2 using CRISPR/Cas9 or overexpression studies. These could be realized in a follow-up project.
- In the “Results” section, several headings are speculative and do not accurately reflect the data presented. For instance, the title “2.1. ENOX2 Protein Expression Serves as a Prognostic Biomarker” is not fully substantiated by the data. The results only demonstrate an association between ENOX2 expression levels and poor survival outcomes, rather than establishing ENOX2 as a definitive prognostic biomarker.
As mentioned in the previous points, we agree with this comment. Therefore, the title of section 2.1 has been changed to “ENOX2 Protein Expression is a Potential Prognostic Marker” and the title of section 2.2 to “ENOX2 RNA Expression is a Potential Prognostic Marker”. As already mentioned, the statements regarding a possible causality of worsened survival due to high ENOX2 expression were interpreted more cautiously, so that only a correlation between the parameters is suggested.
- Additionally, this section lacks sufficient background and context, and includes unnecessary commentary (e.g., “Interestingly, we…”), which should be avoided in objective reporting.
We agree with this comment. Therefore, unnecessary commentary has been removed from the text. The first sentence has been rephrased and moved to a more appropriate place to ensure better understanding (see Section 2.1).
- Furthermore, there is no reference to any figure supporting the starting statement: “Interestingly, we were able to detect immunohistochemical ENOX2 expression in benign cells such as keratinocytes and melanocytes. The staining in all samples indicates an intracellular localization of the ENOX2 protein.” This claim requires proper visual or analytical support, and it would be preferable to include this data as a Supplementary figure.
We agree with this comment. Representative reference images for ENOX2 staining intensities in benign tissue were added in Figure A1.
- Moreover, it is important to comment on the sample sizes presented in Table 1. Specifically, in the tumor “Stage (PM)” category, stage IV includes only three samples: one with low ENOX2 expression and two with high. Drawing a conclusion about ENOX2 expression and tumor progression based on these numbers is not justified.
We agree with this comment. Therefore, a comment on the small number of stage IV patients has been added to the discussion (see line 319-322). This is also one of the reasons why the subgroup with stage I/II was analysed. The statements on metastasis are significant in the group with stage I and II patients. The prognostic information is also more relevant for patients with low tumour stages, as no metastasis has yet occurred, which has a significant influence on therapy.
If you refer to the statement about tumor progression from nevi via primary melanomas to melanoma metastases, the tissue microarray contained a total of 13 melanoma metastases. The three stage IV samples are samples of the primary tumor, not of the metastases.
- In the “Results” section 2.3, titled “Melanoma Cell Lines Mainly Express a 72 kDa ENOX2 Protein”, the title does not accurately reflect the presented findings. It remains unclear whether ENOX2 expression is predominantly observed in melanoma cells. If this is the case, comparisons with other cell types, such as keratinocytes, should be included to substantiate this claim. Alternatively, if the statement refers to comparisons with other proteins, appropriate control experiments are necessary.
To clarify the objectives of this paragraph, a new introductory sentence has been added. Additionally, the section has been moved to a different location and the introduction was modified, to improve understanding (see Section 2.1). This section is not intended to compare ENOX2 expression in melanomas with other cell lines or to compare ENOX2 expression with other proteins, but rather to specify which ENOX2 isoforms could be identified in melanoma cells. Additionally, it demonstrates that, contrary to previous literature, a full-length ENOX2 protein, among others, was found in melanoma cells. Nevertheless, the comparison in the Western blot with other benign cell lines is of interest, which is why we have conducted further experiments with keratinocytes, fibroblasts as well as melanocytes (See Figure A3). Western blot of benign cell lines showed the expression of the same ENOX2 isoform as in melanoma cell lines.
- The “Western Blot” analysis presented in Figure 4 lacks essential controls: 1) A secondary antibody control, where the primary antibody is omitted, is needed to confirm specific binding of the primary antibody. 2) A loading control should be included by stripping and re-blotting against housekeeping proteins such as actin or tubulin.
We agree with this comment. Therefore, the loading control has been added to the Western Blot (see Figure 1). A secondary antibody control is not strictly necessary, as the ENOX2 bands could not be detected when using the same secondary antibody with other primary antibodies and similar exposure times.
A secondary antibody control for the immunohistochemical stainings had already been performed beforehand, which also showed no nonspecific reactions.
- Overall, the text in this section lacks clarity and depth, as it primarily describes western blot bands across different cell lines without drawing meaningful conclusions.
As mentioned before, to clarify the objectives of this paragraph, a new introductory sentence has been added. Additionally, an effort was made to condense the results (see Section 2.1).
- In the “Results” section 2.4, titled “Growth Inhibitory Effects of Phenoxodiol on Melanoma Cells”, providing background information on the treatments used would enhance the reader’s understanding of the results presented. Additionally, each experiment’s objectives should be clarified before listing the values that are already presented in the figures.
Several sentences briefly referencing the methods and objectives of each experiment were added to improve readability. Additionally, background information on the therapies mentioned in the introduction was briefly revisited. Furthermore, an effort was made to condense the results (see Section 2.4).
- Specifically, on page 8, line 172: The reference to “Figure 6a and 2a show…” should be relocated to a new paragraph to improve readability. Furthermore, the text should be enriched with background information to better contextualize the findings.
The reference to the figures was relocated as suggested (see line 178). Background information was added, as previously mentioned.
- Please clarify what Y axis represents in Figure 1.
Y axes represents the relative number of cases. The Y axis labeling has been changed in the figure (see Figure 2).
- Please clarify what “S” and “R” represent in Figure 4.
“S” represents cell lines sensitive to vemurafenib, meanwhile “R” stands for resistance to vemurafenib. Explanations for the abbreviations have been added to the subtitle of Figure 1.
- The “Tissue Microarray” method is described as immunohistochemistry. If immunohistochemistry was indeed used, it would be ideal to include representative images in the supplementary material or appendix.
We agree with this comment. The title of this section was changed to “Immunohistochemistry of the Tissue Micoarray”. Representative reference images for ENOX2 staining intensity were added in Figure A2.
- Additionally, there are issues with the language in this section, where the description starts with the singular "the side" and later shifts to "slides" and "scans," which disrupts clarity.
We agree with this comment. The terminology was adjusted to ensure an accurate description of the method. (line 418, 425 and 427 )
- Furthermore, the statement “only the ENOX2 staining intensity of the melanoma cells/melanocytes was evaluated” requires clarification. Please explain which other cell types were present and how they were excluded during the scanning process.
Other present cell types included cells found in skin biopsies, such as keratinocytes, fibroblasts, or immune cells. A visual identification of the melanoma cells/tumor area was carried out by the evaluator. The staining intensity of ENOX2 in this area was assessed. The description in the methods was revised accordingly (see line 428-429).
- In “Colony Formation Assay”, crystal violet staining stains all cell, including non-colony forming cells. It is unclear how these cells were excluded.
A visual identification of the colonies was performed by the evaluator. Non-colony-forming cells were also excluded/not recorded based on the resolution of the scan. Relevant sections in the methods were revised accordingly (see line 478-479).
- Language
A revision of the language was conducted.
Reviewer 2 Report
Comments and Suggestions for Authors
The authors provide a detailed analysis of Enox2 in melanoma along with combination treatment with BRAF and ENOX inhibitors. Their analysis includes a significant number of patients in a tissue array.
Could the authors provide examples of staining and how the scoring was done on the IHC array? Was this the same antibody that was used for the cell lines? The authors mentioned multiple isoforms that are reported by others. Did the other researchers use the same antibody as the authors? What is the epitope of the antibody? Is it possible that the antibody used does not pick up the isoforms?
Additionally, can specific isoform expressions be checked in the TCGA data?
The Mel1617 cells do not express the 70kd isoform significantly, but the ic20 seems low for Phenoxodiol. Is the drug equally effective for all isoforms and truncated proteins? Is there any correlation between protein levels and ic20? Also, why is IC20 used instead of IC50?
In their combination study, did the authors try to determine the synergy? Or are they showing only combinatorial effects? Phenoxodiol seems quite effective by itself. Why was A375 chosen for the combinatorial effect?
Line 45 ubichinol. Is it spelled correctly?
Author Response
- Could the authors provide examples of staining and how the scoring was done on the IHC array?
We agree with this comment. Representative reference images for ENOX2 staining intensity were added in Figure A2.
- Was this the same antibody that was used for the cell lines? The authors mentioned multiple isoforms that are reported by others. Did the other researchers use the same antibody as the authors? What is the epitope of the antibody? Is it possible that the antibody used does not pick up the isoforms?
The same polyclonal, knock-down validated antibody was used for immunohistochemistry and the western blot of the cell lines. Immunogen for the antibody is the ENOX2 fusion protein Ag0674 (peptide sequence: MIQSANSHVRRLVNEKAAHEKDMEEAKEKFKQALSGILIQFEQIVAVYHSASKQKAWDHFTKAQRKNISVWCKQAEEIRNIHNDELMGIRREEEMEMSDDEIEEMTETKETEESVSQAEALKEENDSLRWQLDAYRNEVELLKQEQGKVHREDDPNKEQQLKLLQQALQGMQQHLLKVQEEYKKKEAELEKLKDDKLQVEKMLENLKEKESCASRLCASNQDSEYPLEKTMNSSPIKSEREALLVGIISTFLHVHPFGASIEYICSYLHRLDNKICTSDVECLMGRLQHTFKQEMTGVGASLEKRWKFCGFEGLKLT (139-455 aa encoded by BC019254). Other researchers used different ENOX2 antibodies, but with similar immunogen peptide sequences:
- PU02 (immunogen: TGVGASLEKRWKFCGFE) or PU04 (immunogen: EEMTECRREEEMEMSDDEIEEMTETK) anti-tNOX antibody (used by Tang et al., 2007)
- anti-ENOX2 single chain variable region of antibody 12.1 (scFv)1 (immunogen: 34-kDa circulating form, Epitopes inhibited the drug-responsive oxidation of NADH with the sera of cancer patients; Cho et al., 2002). As TGVGASL serves as the NADH binding sequence of ENOX2 (Chueh et al., 2002), this could be a potential immunogen of the 12.1 antibody (used by Morre et al., 2008; Hostetler et al., 2009).
Therefore, we cannot tell the exact epitope of the antibody, but because of the use of similar immunogens it seems unlikely that the antibody does not pick up the isoforms. This line of argument was also added to the discussion: „In addition, a different antibody, albeit with a similar immunogen, was used, which could also influence the detection of ENOX2 isoforms.“ (line 237-239)
- Additionally, can specific isoform expressions be checked in the TCGA data?
Unfortunately, specific isoform expression cannot be checked in the TCGA data.
- The Mel1617 cells do not express the 70kd isoform significantly, but the ic20 seems low for Phenoxodiol. Is the drug equally effective for all isoforms and truncated proteins? Is there any correlation between protein levels and ic20? Also, why is IC20 used instead of IC50?
Since the binding site of phenoxodiol is the EEMTE sequence, which also serves as a binding site for ubiquinol and is present in the truncated ENOX2 forms (Chueh et al., 2002), it can be assumed that these are also inhibited by phenoxodiol.
The correlation between IC20 values and ENOX2 expression in the Western blot should be interpreted with caution due to the methodology (e.g., the different cell lines were run on different gels). Therefore, such an analysis was not performed. However, there are indications of a correlation within the sensitive and resistant cell line pairs. Figure 4 was revised to clarify which samples were run on the same gel.
IC20 was used since IC50 was not achieved in all cell lines. A higher inhibitor concentration was not used because inhibition of topoisomerase II has been described at higher ENOX2 concentrations (Constantinou et al., 2002; Porter et al., 2020). Our interest, however, was in the effects mediated by ENOX2 inhibition. This line of argument was also added to the discussion: „Higher doses of PXD were not used in our study to prevent non-ENOX2-mediated effects of PXD, such as inhibition of topoisomerase. Therefore, the IC20 values were used to compare the effects of PXD on the cell lines.“ (line 325-327)
- In their combination study, did the authors try to determine the synergy? Or are they showing only combinatorial effects? Phenoxodiol seems quite effective by itself. Why was A375 chosen for the combinatorial effect?
The data do not indicate synergy; therefore, no synergy analysis was conducted. Additionally, as previously mentioned, no resensitization of the vemurafenib-resistant cells under phenoxodiol was observed.
A375 was chosen as a representative cell line. The results for the other cell lines can be found in the appendix.
- Line 45 ubichinol. Is it spelled correctly?
We agree with this comment and corrected to „ubiquinol“. (line 46)